# Image Fusion for High-Resolution Optical Satellites Based on Panchromatic Spectral Decomposition

**DOI:** 10.3390/s19112619

**Published:** 2019-06-09

**Authors:** Luxiao He, Mi Wang, Ying Zhu, Xueli Chang, Xiaoxiao Feng

**Affiliations:** 1State Key Laboratory of Information Engineering in Surveying, Mapping and Remote Sensing (LIESMARS), Wuhan University, Wuhan 430079, China; heluxiao@whu.edu.cn (L.H.); yzhu_1003@foxmail.com (Y.Z.); fengxx2018@whu.edu.cn (X.F.); 2Collaborative Innovation Center of Geospatial Technology, 129 Luoyu Road, Wuhan 430079, China; xl_whu@foxmail.com

**Keywords:** ratio transformation, panchromatic spectral decomposition, panchromatic and multi-spectral fusion, high-resolution optical satellites

## Abstract

Ratio transformation methods are widely used for image fusion of high-resolution optical satellites. The premise for the use the ratio transformation is that there is a zero-bias linear relationship between the panchromatic band and the corresponding multi-spectral bands. However, there are bias terms and residual terms with large values in reality, depending on the sensors, the response spectral ranges, and the land-cover types. To address this problem, this paper proposes a panchromatic and multi-spectral image fusion method based on the panchromatic spectral decomposition (PSD). The low-resolution panchromatic and multi-spectral images are used to solve the proportionality coefficients, the bias coefficients, and the residual matrixes. These coefficients are substituted into the high-resolution panchromatic band and decompose it into the high-resolution multi-spectral bands. The experiments show that this method can make the fused image acquire high color fidelity and sharpness, it is robust to different sensors and features, and it can be applied to the panchromatic and multi-spectral fusion of high-resolution optical satellites.

## 1. Introduction

In the conflict between spectral resolution and spatial resolution, the existing optical satellites generally provide high-resolution panchromatic images but low-resolution multi-spectral images [1]. Panchromatic images provide the overall spatial structure information, which can describe the structural details of features. The multi-spectral images provide spectral information of the features, which facilitates feature recognition, as well as their classification and interpretation. The spatial and spectral structures are both important components of the remote-sensing information. The panchromatic and multi-spectral image fusion technology can combine the advantages of both components to obtain high-resolution multi-spectral images [2].

Fusion methods for panchromatic and multi-spectral images can be roughly divided into coefficient enhancement methods and component transformation methods. The coefficient enhancement methods sharpen the multi-spectral bands through the panchromatic band. The typical methods include Brovey [3], SFIM (smoothing filter-based intensity modulation) [4], SVR (synthetic variable ratio) [5], UNB_Pansharp (University of New Brunswick Pansharp) [6], etc. The Brovey algorithm uses a part of the panchromatic image as a multi-spectral band, which is simple and easy to understand. However, the fusion results may suffer the problem of serious color distortion. The methods of SFIM, SVR, and UNB_Pansharp all have the same fusion formula to sharpen multi-spectral images by the ratio of the high-resolution panchromatic image to the low-resolution panchromatic image. Therefore, these methods can be described as ratio transformation methods. The difference among these three methods is how to generate the low-resolution panchromatic image, and how to optimize the sharpness and maintain the original color and spectrum [7]. The SFIM algorithm uses the smoothing filter to generate the low-resolution panchromatic image and further obtains modulation coefficients. The size, type, and parameters of convolution kernel will affect the final fusion results. Moreover, SFIM has the problem of low sharpness and distortion of ground details. The SVR algorithm uses the least squares method (LSM) [8] to calculate the contribution of each band of the multi-spectral image to the panchromatic image, and weights these bands to obtain the low-resolution panchromatic image. The UNB_Pansharp algorithm is a further optimization of SVR to restrain the color distortion and improve sharpness by statistical methods. Since the goal of UNB_Pansharp is to achieve global optimization, there will be local color distortion for images of large size or complex features.

The component transformation methods transform the multi-spectral image into another space, replace one of the components with the panchromatic image, and perform an inverse transform to obtain the final fused image. Typical methods include HIS (hue, intensity, saturation) [9], GS (Gram–Schmidt) [10], PCA (principal components analysis) [11], etc. The HIS algorithm converts the RGB (red, green, blue) color space into the HIS color space, and replaces the intensity component with the panchromatic image. The HIS is only suitable for images with three bands, and also has a color distortion problem. The GS algorithm transforms the image orthogonally and replaces the first component with the panchromatic image. The GS algorithm has high complexity which is not suitable for large-size image processing. In addition, there are many other fusion methods based on wavelet transform [12], Laplacian pyramid [13], contourlet transform [14], sparse coding [15], etc.

For high-resolution remote-sensing images, most fusion methods will cause color distortion [16]. Relatively speaking, UNB_Pansharp, GS, and SFIM have better color retention capabilities, but still have local color distortion, high algorithm complexity, feature distortion, and other shortcomings [17]. Therefore, a fusion algorithm with low color distortion is still worth studying.

This paper firstly analyzes the ratio transformation methods, and points out that a zero-bias relationship between panchromatic and multi-spectral images is the premise of these methods. However, it is inaccurate to describe the two with a zero-bias linear relationship. There are bias and residual terms between the two in practice. Aiming at this imperfection, this paper proposes a fusion method called panchromatic spectral decomposition (the PSD algorithm). This method can convert a panchromatic image into every high-resolution multi-spectral band directly by constructing the relationship between the two. The algorithm characteristics and performance are tested using simulated data. The algorithm stability is further tested with four sets of real data from different sensors. Experiments show that the PSD algorithm allows fused images to have low color distortion, and is robust to different high-resolution optical sensors.

## 2. Methodology

### 2.1. Relationship between Panchromatic and Multi-Spectral Pixels

The ratio transformation methods belong to the coefficient enhancement methods, which are commonly used in the field of high-resolution optical remote sensing. The theoretical basis is that there is a strong correlation between the panchromatic and the multi-spectral images for the simultaneous phase [18]. The ratio transformation methods consider that the ratio of the original panchromatic image to the degraded panchromatic image is equal to the ratio of the fused image to the up-sampled multi-spectral image [19], which can be described as follows:(1)$$\frac{Pan_{HQ}(i,j)}{Pan_{LQ}(i,j)}=\frac{MS_{HQ}^{\left(g\right)}(i,j)}{MS_{LQ}^{\left(g\right)}(i,j)},$$
where $Pan_{HQ}$ is the high-quality panchromatic image (original panchromatic image), $Pan_{LQ}$ is the low-quality panchromatic (degraded panchromatic image), $MS_{HQ}$ is the high-quality multi-spectral image (fused image), $MS_{LQ}$ is the low-quality multi-spectral image (up-sampled multi-spectral image), and i, j, and g are the indices of the row, column, and band, respectively.

The ratio transformation methods are devoted to construct a reasonable $Pan_{LQ}$. It can be obtained in two ways. One approach is band weighting of the multi-spectral bands, such as UNB_Pansharp and its improved methods [20]. The other approach is panchromatic image degrading, such as SFIM and its improved methods [21].

The correctness of Equation (1) is based on the idea that there is zero-bias linear relationship between panchromatic and multi-spectral images. However, this premise is not accurate enough in practice. A set of Beijing-2 data are taken as an example. Beijing-2 has a panchromatic band (Pan: 450–650 nm) and four multi-spectral bands (B1: 440–510 nm, B2: 510–590nm, B3: 600–670 nm, B4: 760–910 nm). Figure 1 records the digital number (DN) distribution of the panchromatic band and its corresponding multi-spectral bands. The distribution of bands 1–3 is more linear than band 4. That is because the spectral range of bands 1–3 has a certain overlap with the panchromatic band, but band 4 does not. It can be seen from Figure 1 that the panchromatic and multi-spectral pixels have correlation, but not a zero-bias linear relationship. Therefore, the relationship between the panchromatic image and a single multi-spectral band is modified as
(2)$$Pan=k^{\left(g\right)}MS^{\left(g\right)}+b^{\left(g\right)}T+{Ε}^{\left(g\right)},$$
where superscript (g) denotes the index of band, Pan is the panchromatic image, $MS^{\left(g\right)}$ is the multi-spectral band, $k^{\left(g\right)}$ is the proportionality coefficient, $b^{\left(g\right)}$ is the bias coefficient, T is a matrix which has same size as Pan and all elements are 1, and $E^{\left(g\right)}$ is the residual matrix. From Equation (2), a single multi-spectral band with the same resolution can be decomposed from the panchromatic image as follows:(3)$$MS^{\left(g\right)}=\frac{1}{k^{\left(g\right)}}(Pan-b^{\left(g\right)}T-E^{\left(g\right)}).$$

Compared with the existing ratio transformation methods, the PSD algorithm adds the bias term and the residual term into the Pan–MS relationship model, which enhances the accuracy. In addition, the ratio transformation methods will add the same high-frequency information into different bands, which is an important cause of color distortion. The PSD algorithm generates the fused image by converting the panchromatic image directly, which can avoid this problem effectively.

### 2.2. Linear Fitting of Panchromatic and Multi-Spectral Bands

The proportionality coefficient $k^{\left(g\right)}$ and the bias coefficient $b^{\left(g\right)}$ reflect the overall linear relationship between the panchromatic band and the corresponding multi-spectral band. The relationship is related to only three factors: (1) the response spectra of the panchromatic band and the corresponding multi-spectral band, (2) the land-cover types and their proportion in the image, and (3) the solar irradiance during imaging. The image degradation model [22] does not include the above three factors, which means that the resolution differences will not change the proportionality and the bias coefficients. Therefore, $k^{\left(g\right)}$ and $b^{\left(g\right)}$ can be solved with the low-resolution bands.

The original panchromatic image ($Pan_{HR}$) needs to be down-sampled to match the size of the original multi-spectral image ($MS_{LR}$). In order to reduce the high-pass effect of down-sampling, $Pan_{HR}$ is blurred first. Since the general Pan–MS resolution ratio is 1:4, the mean filter of 5×5 is chosen to blur $Pan_{HR}$. Subsequently, $Pan_{HR}$ is down-sampled to get $Pan_{LR}$. After these processes, the pixels with the same positions of $Pan_{LR}$ and $MS_{LR}$ are constituted into a set of valid samples.

Due to the spatial correlation of remote-sensing images, there is redundant information between adjacent pixels. Having all the pixels participating in the fit will greatly increase the cost of the calculation. Therefore, samples are taken from every 10th row and every 10th column. In addition, the DN value of standard remote-sensing data has a certain numerical limit. For example, Beijing-2 is stored in 10 bits, and its maximum DN value is 1023, which means that the overexposure objects will have the value of 1023. These overexposure pixels do not conform to the linear relationship described in Equation (2); thus, they are treated as gross errors and need to be rejected. After collecting the valid samples, the proportionality and the bias coefficients can be fitted by LSM [8].

### 2.3. Estimation of the Residual Matrix

Based on the above analyses, the panchromatic and multi-spectral bands with different resolution have relationships as follows:(4)$$\{\begin{array}{c} Pan_{HR}=k^{\left(g\right)}MS_{HR}^{\left(g\right)}+b^{\left(g\right)}T+{Ε}_{HR}^{\left(g\right)}\\ Pan_{LR}=k^{\left(g\right)}MS_{LR}^{\left(g\right)}+b^{\left(g\right)}T+{Ε}_{LR}^{\left(g\right)} \end{array}.$$

If ${Ε}_{HR}^{\left(g\right)}$ is replaced by the up-sampled ${Ε}_{LR}^{\left(g\right)}$ (written as ${Ε}_{LR}^{\left(g\right)}↑$), then the ideal high-resolution multi-spectral band ($MS_{HR}^{\left(g\right)}$) can be expressed as
(5)$$MS_{HR}^{\left(g\right)}=\frac{1}{k^{\left(g\right)}}(Pan_{HR}-b^{\left(g\right)}T-{Ε}_{LR}^{\left(g\right)}↑)-\frac{1}{k^{\left(g\right)}}({Ε}_{HR}^{\left(g\right)}-{Ε}_{LR}^{\left(g\right)}↑).$$

Therefore, the fused image (Fusion) can be expressed as
(6)$$Fusion^{\left(g\right)}=\frac{1}{k^{\left(g\right)}}(Pan_{HR}-b^{\left(g\right)}T-{Ε}_{LR}^{\left(g\right)}↑)=MS_{HR}^{\left(g\right)}+\Delta ,$$
where $\Delta =\frac{1}{k^{\left(g\right)}}({Ε}_{HR}^{\left(g\right)}-{Ε}_{LR}^{\left(g\right)}↑)$ is the error matrix.

There will be an error matrix Δ in the fused image if ${Ε}_{HR}^{\left(g\right)}$ is replaced by ${Ε}_{LR}^{\left(g\right)}↑$. No matter what the resolution is, the residual matrix E can be divided into two parts. One part will cause color distortion, and the other part will cause spatial distortion. Since ${Ε}_{HR}^{\left(g\right)}$ and ${Ε}_{LR}^{\left(g\right)}↑$ are only different in original resolution, the color distortion effects caused by them are the same, and can be canceled out by ${Ε}_{HR}^{\left(g\right)}-{Ε}_{LR}^{\left(g\right)}↑$. Meanwhile, ${Ε}_{HR}^{\left(g\right)}$ contains the spatial information of ${Ε}_{LR}^{\left(g\right)}↑$ completely, so ${Ε}_{HR}^{\left(g\right)}-{Ε}_{LR}^{\left(g\right)}↑$ will retain some high-frequency spatial information. The above analyses illustrate that the existence of Δ will cause spatial distortion, but will not cause color distortion. The spatial distortion appears to be jagged in detail, especially in high-gradient areas. This is due to the fact that $E_{LR}↑$ is not smooth enough following the up-sampling. In order to avoid or weaken this spatial distortion, we perform smooth filtering on $E_{LR}↑$ to make the edges smoother. Above all, the final fused image is expressed as
(7)$$Fusion^{\left(g\right)}=\frac{1}{k^{\left(g\right)}}(Pan_{HR}-b^{\left(g\right)}T-{Ε}_{LR}^{\left(g\right)}↑⊗h),$$
where the convolution kernel h used here is the mean filter kernel of 3×3.

### 2.4. Constraints on Extreme Values

Saturated pixels do not reflect the true spectral information of the objects. In the fitting process, the saturated pixels are already excluded. However, in the process of panchromatic band decomposition, it is not considered whether a pixel is saturated. In this case, errors easily occur in high-DN areas, which will enlarge the dynamic range of the fused image and cause global color deviation. To avoid this, the color distortion caused by dynamic range change can be attenuated by some extreme value constraint. The up-sampled multi-spectral band ($MS_{LR}↑$) is treated as standard, whereby the maximum value of each line in the fused image cannot be larger than the maximum value of the corresponding line in the multi-spectral image, and the minimum value of each line in the fused image cannot be less than the minimum value of the corresponding line in the multi-spectral image. The constraint is described as
(8)$$\{\begin{array}{l} Fusion^{\left(g\right)}(i,j)=\min[MS_{LR}^{\left(g\right)}↑(i,:)]\;if\;Fusion^{\left(g\right)}(i,j)<\min[MS_{LR}^{\left(g\right)}↑(i,:)]\\ Fusion^{\left(g\right)}(i,j)=\max[MS_{LR}^{\left(g\right)}↑(i,:)]\;if\;Fusion^{\left(g\right)}(i,j)>\max[MS_{LR}^{\left(g\right)}↑(i,:)] \end{array},$$
where $MS_{LR}^{\left(g\right)}↑(i,:)$ denotes the ith line of $MS_{LR}^{\left(g\right)}↑$, min[·] denotes getting the minimum value, and max[·] denotes getting the maximum value.

### 2.5. Summary of the PSD Algorithm

The PSD algorithm is summarized in Figure 2, which has five steps: (1) mean filtering and down-sampling the original panchromatic image to obtain the low-resolution panchromatic image with the same size as the original multi-spectral image; (2) obtaining the proportionality coefficient, the bias coefficient, and the low-resolution residual matrix by linear fitting of the low-resolution panchromatic image and the original multi-spectral image; (3) up-sampling and mean filtering the low-resolution residual matrix to replace the high-resolution residual matrix; (4) decomposing the panchromatic image into high-resolution multi-spectral bands; and (5) constraining the extreme value to obtain the final fused image.

## 3. Experiments and Discussion

### 3.1. Experimental Data

Many quality metrics require a high-resolution multi-spectral image as a reference image [23]. Based on this consideration, the experiments are divided into two parts: simulation data experiments and real data experiments. The simulation data are used for linear fitting analysis, residual matrix analysis, and comparison of different algorithms. The real data are used for algorithm stability analysis to test the stability of PSD algorithm for different sensors.

In simulation data experiments, a set of down-sampled Beijing-2 panchromatic and multi-spectral images are used as simulation data for quantitative analysis, to ensure the reliability of the conclusions. Beijing-2 is a constellation constituted by three push-broom optical remote-sensing satellites, providing panchromatic images with 0.8-m resolution (Pan: 450–650 nm) and multi-spectral images (B1: 440–510 nm, B2: 510–590 nm, B3: 600–670 nm, B4: 760–910 nm). The study area is Kunming, China, with coordinates of 102.6176 east (E) and 25.1246 north (N). The size of the original panchromatic image is 6000×6000, and the size of the original multi-spectral image is 1500×1500. Down-sampled by Gaussian pyramid [24], the size of the simulated panchromatic image is 1500×1500, and the size of simulated multi-spectral image is 375×375. This simulation data can produce a fused image with size of 1500×1500. Taking the original multi-spectral image as the reference image, the fusion result can be quantitatively evaluated.

In real data experiments, this paper selects Beijing-2, SkySat-3, SuperView-1, and ZiYuan-3 as experimental data. The experimental data cover various features such as city, farmland, forest, and mountain. The specific conditions are listed in Table 1.

### 3.2. Analysis of Linear Fitting

The simulated Beijing-2 data are applied for linear fitting analysis. Figure 3 records the relationship between the panchromatic band and multi-spectral band 1 before and after the residual matrix elimination. Table 2 records the linear fitting results include proportionality coefficient, bias coefficient, determination coefficients (*R*^2^), and root-mean-square error (RMSE) [25]. The *R*^2^ and RMSE are used to measure the goodness of fit [26]. Higher *R*^2^ and lower RMSE indicate better fitting goodness.

It can be seen from Figure 3 and Table 2 that, after subtracting the residual matrix from the panchromatic image, the DN value will obviously move closer to the fitted straight line. The all-fitting goodness of different bands is improved, especially band 4 (near-infrared, NIR). In addition, the fitting parameters of different bands are basically unchanged before and after the residual matrix elimination. This experiment can prove that the linear relationships are the same for different resolutions. Therefore, it is reasonable to use the low-resolution panchromatic and multi-spectral bands to solve the proportionality and bias coefficients.

### 3.3. Analysis of the Residual Matrix

The simulated Beijing-2 data are applied for residual matrix analysis. Figure 4 presents a set of visual residual matrices, where (a) is the high-resolution residual matrix ($E_{HR}$), (b) is the up-sampled low-resolution residual matrix ($E_{LR}↑$), and (c) is the difference between $E_{HR}$ and $E_{LR}↑$ ($E_{HR}-E_{LR}↑$). The root mean square (RMS) is applied to indicate the intensity of the residual matrix, and the RMS values of the above three are recorded in Table 3.

From the comparison of Figure 4 and Table 3, the following conclusions can be drawn: (1) $E_{HR}$ and $E_{LR}↑$ have similar color information, and $E_{HR}-E_{LR}↑$ basically only retains some high-frequency spatial information; (2) the information of $E_{LR}↑$ is completely included in $E_{HR}$, so the residual intensity of $E_{HR}$ is larger than $E_{LR}↑$; (3) the related parts of $E_{HR}$ and $E_{LR}↑$ can be eliminated, so the residual intensity of $E_{HR}-E_{LR}↑$ is less than $E_{HR}$.

### 3.4. Comparison of Different Algorithms

The Beijing-2 simulation data are used to test the performance of the proposed fusion algorithm. The comparison methods include Brovey, GS, PCA, SFIM, and UNB_Pansharp, where Brovey, GS, and PCA are algorithm modules in ENVI (The Environment for Visualizing Images), and UNB_Pansharp is an algorithm module in PCI Geomatica. The results are shown in Figure 5, and the corresponding evaluation parameters are given in Table 4. The quality metrics include RMSE, signal-to-noise ratio (SNR), relative dimensionless global error in synthesis (ERGAS) [27], and correlation coefficient (CC) [28]. The performance of the algorithms is compared by the similarity of the fused image to the reference image. If an algorithm performs well, then RMSE and ERGAS will have smaller values, while SNR and CC will have larger values.

Figure 5 shows that there is no serious color distortion in any of the fusion images. The PSD, Brovey, and SFIM have the closest color performance, while the shadows in PCA and GS have some distortion, the color of SFIM and UNB_Pansharp tend toward blue, and the color of the roof in UNB_Pansharp is too dark. In terms of the image sharpness, the UNB_Pansharp has the best performance, and the PSD and SFIM are second to it. In terms of the feature details, no visible structural variation occurred except for SFIM. In the fused image of SFIM, there is a phenomenon of intermittent lines on the highway. This is because the low-resolution panchromatic of SFIM is generated by smooth filtering, which does not work well with linear objects. Compared with the other algorithms, PSD has the best visual effect in Figure 5.

Brovey can only be applied for three bands; thus, the mean values of the quality metrics are calculated from band 1 to band 3. From the parameters in Table 4, PSD has the best performance, and PCA and SFIM are second, followed by UNB_Pansharp, whereas Brovey and GS are the worst. We note that CC sometimes deviated from the other three metrics. For example, Brovey has bad performance in RMSE, SNR, and ERGAS, but good performance in CC. This is because RMSE, SNR, and ERGAS measure the absolute difference between the fused image and the reference image, while CC measures the relative difference between the two.

The PSD algorithm still has some shortcomings. Compared with the other three bands, band 4 of the PSD fused image does not perform well. It can be seen from Figure 1, Table 2, and Table 4 that the performance of PSD depends on the correlation between the panchromatic and multi-spectral bands. Since the spectra of the Beijing-2 panchromatic band and multi-spectral band 4 do not overlap, the correlation between the two is relatively low.

### 3.5. Analysis of Algorithm Stability

Figure 6 records four sets of fusion results, where (a1), (b1), (c1), and (d1) are the up-sampled multi-spectral images, and (a2), (b2), (c2), and (d2) are the corresponding fused images. Table 5 records their corresponding evaluation parameters.

Through the fusion results of the four different data sets, we can see in Figure 6 that the colors of the fused images are consistent with their corresponding up-sampled multi-spectral images, and no significant color distortion occurs. Meantime, the fused images are sharp enough for visual effects, and there is no obvious spatial structure deformation.

There are no real high-resolution multi-spectral images; thus, the up-sampled multi-spectral images are used as reference images to calculate evaluation parameters. However, this operation will make the increased high-frequency information in fused images be treated as noise. In this case, the evaluation parameters will be relatively worse than normal. For example, the evaluation parameters of the real Beijing-2 fused image are worse than parameters of the simulated Beijing-2 fused image. Despite such unfavorable conditions, the four sets of data still achieve good performance, where the RMSE is maintained between 13.62 and 28.5, SNR is maintained between 15.57 and 23.00, CC is maintained between 0.81 and 0.96, and ERGAS is maintained between 2.21 and 4.89. In summary, the PSD algorithm is robust for different types of satellites and sensors, and it is suitable for the panchromatic and multi-spectral fusion of high-resolution optical satellites.

## 4. Conclusions

This paper proposes a fusion method which can directly convert the panchromatic band into the high-resolution multi-spectral band by constructing the relationship between the two. Experiments show that the proposed method has good color retention ability. Compared with UNB_Pansharp, RMSE decreased from 13.82 to 7.16, ERGAS decreased from 4.89 to 2.54, SNR increased from 15.41 to 21.13, and CC increased from 0.91 to 0.98. The proposed method has good adaptability to different sensors and features by testing multiple sets of data. The current problem of this algorithm is that the fusion performance depends on the correlation between the panchromatic band and the multi-spectral band. Bands 1–3 have strong correlation with the panchromatic band, and the RMSEs of their fusion results are 6.73, 6.59, and 8.17, respectively. Band 4 has a weak correlation with the panchromatic band, and the RMSE is increased to 18.94. How to improve the algorithm’s adaptation to Pan–MS correlation is a worthy direction in future work.

## Figures and Tables

**Figure 1 sensors-19-02619-f001:**
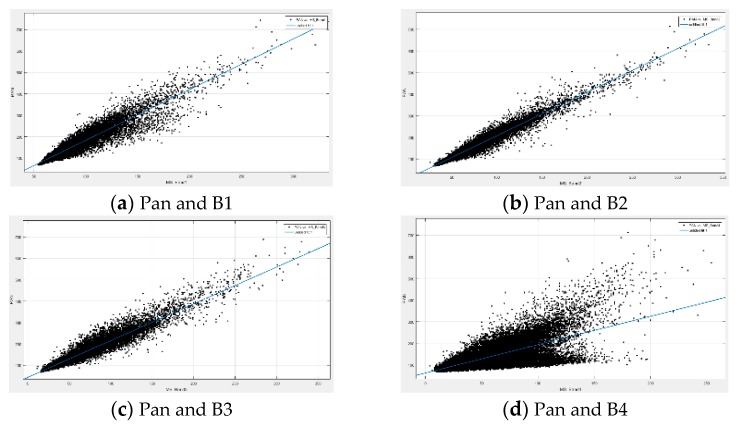
Digital number (DN) relationships between the panchromatic band and the different multi-spectral bands, where (**a**), (**b**), (**c**), and (**d**) are the distribution of the panchromatic band and multi-spectral bands 1–4, respectively.

**Figure 2 sensors-19-02619-f002:**
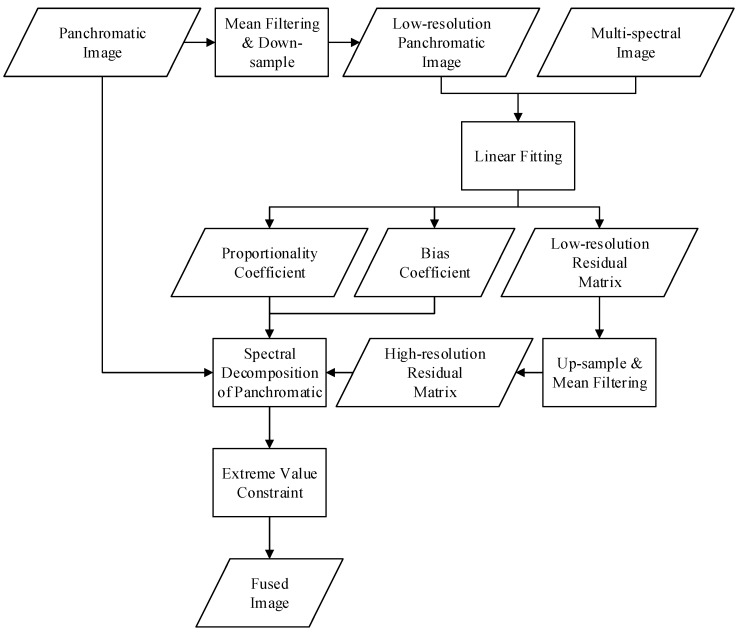
The flow chart of the panchromatic spectral decomposition (PSD) algorithm.

**Figure 3 sensors-19-02619-f003:**
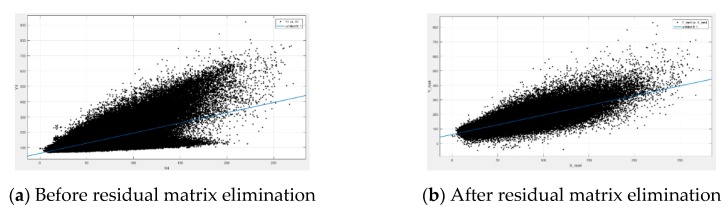
The DN distribution of the panchromatic image and multi-spectral band 1, where (**a**) is the distribution before residual matrix elimination, and (**b**) is the distribution after residual matrix elimination.

**Figure 4 sensors-19-02619-f004:**
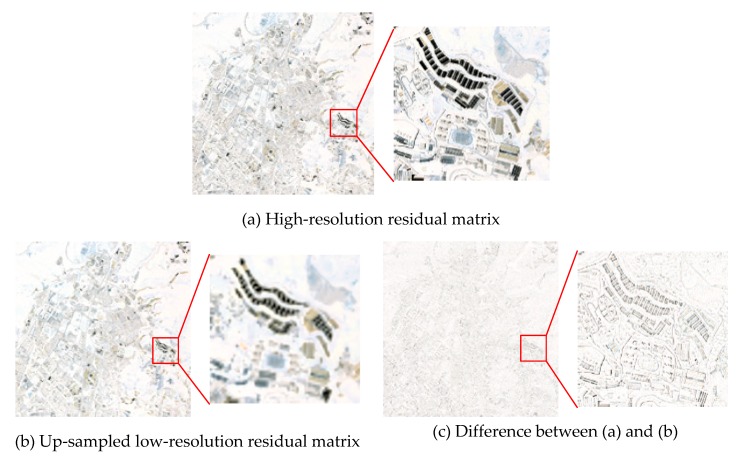
Visual residual matrix after linear fitting, where (**a**) is the high-resolution residual matrix, written as $E_{HR}$, (**b**) is the up-sampled low-resolution residual matrix, written as $E_{LR}↑$, and (**c**) is the difference between (**a**) and (**b**), written as $E_{HR}-E_{LR}↑$.

**Figure 5 sensors-19-02619-f005:**
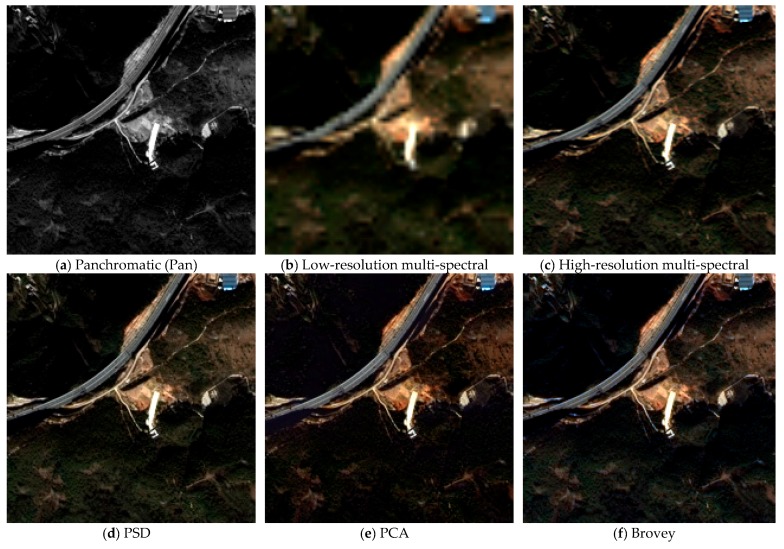
Fusion results produced by different algorithms, where (**a**) is the panchromatic image, (**b**) is the low-resolution multi-spectral image, (**c**) is the high-resolution multi-spectral image, (**d**) is the fused image produced by PSD, (**e**) is the fused image produced by principal component analysis (PCA), (**f**) is the fused image produced by Brovey, (**g**) is the fused image produced by Gram-Schmidt (GS), (**h**) is the fused image produced by smoothing filter-based intensity modulation (SFIM), and (**i**) is the fused image produced by UNB_Pansharp.

**Figure 6 sensors-19-02619-f006:**
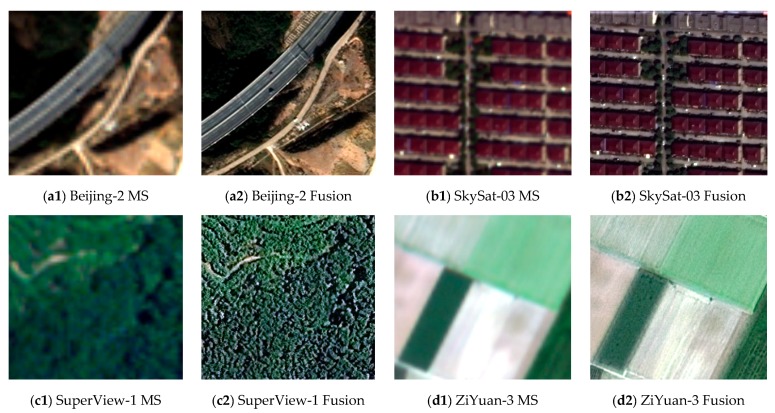
Fusion results of different data, where (**a1**) is the up-sampled multi-spectral image of Beijing-2, (**a2**) is fused image of Beijing-2, (**b1**) is the up-sampled multi-spectral image of SkySat-03, (**b2**) is the fused image of SkySat-03, (**c1**) is the up-sampled multi-spectral image of SuperView-1, (**c2**) is the fused image of SuperView-1, (**d1**) is the up-sampled multi-spectral image of ZiYuan-3, and (**d2**) is the fused image of ZiYuan-3.

**Table 1 sensors-19-02619-t001:** Conditions of experimental data. ID—identifier; Pan—panchromatic; MS—multi-spectral.

Satellite ID	Resolution of Pan (m)	Resolution of MS (m)	Band Number	Feature Types
Beijing-2	0.8	3.2	4	City and mountain
SkySat-03	0.9	2	4	City and farmland
SuperViwe-1	0.5	2	4	Forest
ZiYuan-3	2.1	5.8	4	Farmland

**Table 2 sensors-19-02619-t002:** Fitting results before and after the residual matrix elimination. RMSE—root-mean-square error.

		k	b	*R* ^2^	RMSE
Band 1	Before	2.365	−53.32	0.8956	21.97
	After	2.378	−54.55	0.9433	15.86
Band 2	Before	2.017	5.958	0.9571	14.09
	After	2.022	5.434	0.9623	13.20
Band 3	Before	1.725	43.35	0.929	18.12
	After	1.723	43.42	0.9561	14.02
Band 4	Before	1.329	61.55	0.3925	52.99
	After	1.34	61.33	0.7176	26.93

**Table 3 sensors-19-02619-t003:** The root mean square (RMS) of residual matrices.

	Band 1	Band 2	Band 3	Band 4	Mean
$E_{HR}$	21.97	14.10	18.15	53.11	26.83
$E_{LR}↑$	13.75	5.07	10.62	42.66	18.02
$E_{HR}-E_{LR}↑$	15.78	13.18	14.06	26.97	17.50

**Table 4 sensors-19-02619-t004:** The evaluation parameters of different fusion results. SNR—signal-to-noise ratio; CC—correlation coefficient; ERGAS—relative dimensionless global error in synthesis; GS—Gram–Schmidt; PCA—principal component analysis; PSD—panchromatic spectral decomposition; SFIM—smoothing filter-based intensity modulation.

Quality Metrics		Brovey	GS	PCA	PSD	SFIM	UNB_Pansharp
**RMSE**	Band 1	29.61	20.99	7.73	6.73	8.13	12.55
	Band 2	26.03	18.35	9.11	6.59	9.64	13.66
	Band 3	25.00	16.20	10.86	8.17	11.41	15.27
	Band 4	-	14.46	13.99	18.94	14.32	17.12
	Mean of RGB	26.88	18.51	9.23	7.16	9.73	13.82
**SNR**	Band 1	9.39	8.13	21.43	22.64	20.99	17.22
	Band 2	9.72	9.64	18.84	21.66	18.35	15.33
	Band 3	9.77	11.41	16.63	19.11	16.20	13.67
	Band 4	-	14.32	14.66	12.03	14.46	12.90
	Mean of RGB	9.63	9.73	18.97	21.13	18.51	15.41
**CC**	Band 1	0.95	0.96	0.97	0.97	0.96	0.89
	Band 2	0.98	0.96	0.97	0.98	0.96	0.91
	Band 3	0.98	0.96	0.97	0.98	0.96	0.92
	Band 4	-	0.92	0.92	0.85	0.92	0.85
	Mean of RGB	0.97	0.96	0.97	0.98	0.96	0.91
**ERGAS**		9.14	3.51	3.33	2.54	3.43	4.89

**Table 5 sensors-19-02619-t005:** The evaluation parameters of different data.

Quality Metrics	Beijing-2	SkySat-03	SuperView-1	ZiYuan-3
**RMSE**	13.62	26.16	28.50	18.38
**SNR**	15.57	22.09	23.00	19.39
**CC**	0.93	0.96	0.81	0.88
**ERGAS**	4.89	3.80	2.21	4.63

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
