# Peer review of "Image Fusion for High-Resolution Optical Satellites Based on Panchromatic Spectral Decomposition"

_sensors, 2019, doi:10.3390/s19112619_

Reviewer 1 Report

This paper is an extension of the UNB-pansharp algorithm in which the authors established a linear regression between pan and MS bands to perform image fusion. However, the contribution of the paper is not clearly described. Plus, the paper does not qualify for publication at this stage. The paper includes a lot of unnecessary formula which not only doesn’t help with understanding the work but also can confuse an average-level reader. I recommend the paper to be overhauled highlighting the main contribution of the worked. The paper also suffers from many grammatical errors and I recommend the authors to get the paper proof read before submitting the next revision.

Below, the authors can find my comments for up to section 4. I didn’t continue reading the paper as I need a clear version first.

L92: replace “serial numbers” with “indices”

L105: it is awkward to have a numerical example as a formula. Move the numerical one to the caption and replace it with a parametric formula.

L112: “but the zero-bias linear relationship does not.” Wrong wording. Rephrase

L113: with “a” large value…

L116: What is the benefit of multiplying a vector by a matrix of ones? Why can’t you just say pan=k*ms+T+E?

L117-119 rephrase

L121: what is the problem of eq 5 that makes you propose a new solution? Please mention here.

L146: Does P stand for the Pan band?

L145: eq 6 is unnecessary and in fact more confusing than helpful. Everybody knows what a mean filter is. Please replace it by just mentioning that you applied a mean filter. Why do you select a 5*5 mean filter in eq. 7? Explain the reason.

L160 to 168 go to section 4 not in methodology.

L170: what is betta? What is b? what is k?

L174:183: the equations are all redundant. We understand you want to do least square. There is no need to elongate the paper with repetitive formula.

L227: up to this line, I still don’t understand what the contribution of the paper is. Using least square for image fusion is well studied before. What is the novelty you bring in? From what I understood, you get the coefficients once from LR and once from HR scenarios. Then you subtract their difference from Pan? Is that the contribution of the paper?

L253-255: The 254 larger the R-square is, the smaller the RMSE is, and the closer the fitting result is…

I stopped reviewing the paper at this stage as I need a clearer version of the paper to continue reviewing.

 Author Response

Dear Reviewer 1:

Thanks for your valuable advices, we have made the following changes based on your suggestions.

 1.       The unnecessary formulas are deleted, form the original 29 to 8.

 2.        The methodology, experiments and discussions have been rewritten, and some language errors have been fixed.

 The follows are responses for your specific questions:

Q1: L92: replace “serial numbers” with “indices”

A1: Modified as required.

 Q2: L105: it is awkward to have a numerical example as a formula. Move the numerical one to the caption and replace it with a parametric formula.

A2: The original formula (3) has been deleted.

 Q3: L112: “but the zero-bias linear relationship does not.” Wrong wording. Rephrase

A3: This sentence is modified as “It can be seen from Figure 1 that the panchromatic and multi-spectral pixels have correlation, but not zero-bias linear relationship.”

 Q4: L113: with “a” large value…

A4: This part has been deleted.

 Q5: L116: What is the benefit of multiplying a vector by a matrix of ones? Why can’t you just say pan=k*ms+T+E?

A5: In original formula 4 (present formula 2), k_(g), b_(g) are both numbers, not vectors, denotes the proportionality and bias coefficients of gth band. Pan, MS_(g), and E_(g) are matrixes, so b_(g) needs to multiply a matrix of ones.

 Q6: L117-119 rephrase

A6: The original L117-119 are modified as L104-106 in current version:

 “Therefore, the relationship between panchromatic image and a single multi-spectral band is modified as: …

Q7: L121: what is the problem of eq 5 that makes you propose a new solution? Please mention here.

A7: The explanation is added as:

“Unlike NUB_Pansharp, the PSD algorithm attempts to construct the relationship between a panchromatic band and a single multi-spectral band, thereby directly converting the panchromatic image into the multi-spectral band. But NUB_Pansharp attempts to construct the relationship between a panchromatic band and multiple multi-spectral bands, synthesizing degraded panchromatic image using multi-spectral bands, and enhance the spatial information by coefficient modulation.

Q8: L146: Does P stand for the Pan band?

A8: The original formula 6 has been deleted, and replaced as:

“The image degradation model [22] does not include the above three factors, which means that the resolution differences will not change the proportionality and the bias coefficients.”

 Q9: L145: eq 6 is unnecessary and in fact more confusing than helpful. Everybody knows what a mean filter is. Please replace it by just mentioning that you applied a mean filter. Why do you select a 5*5 mean filter in eq. 7? Explain the reason.

A9: The original formula 6 and formula 7 have been deleted, and the description is modified as:

“In order to reduce the high-pass effect of down-sampling, Pan_HR will be blurred first. Since the general Pan-MS resolution ratio is 1:4, the mean filter of 5*5 is chosen to blur Pan_HR.”

 Q10:L160 to 168 go to section 4 not in methodology.

A11: This part is used to describe how to obtain valid samples and improve efficiency, which is part of the methodology. So this part is still in the text.

 Q11: L170: what is betta? What is b? what is k?

A11: The specific description about LSM has been deleted, and replaced by a reference paper.

 Q12: L174:183: the equations are all redundant. We understand you want to do least square. There is no need to elongate the paper with repetitive formula.

A12: The original formula (12) has been deleted.

 Q13: L227: up to this line, I still don’t understand what the contribution of the paper is. Using least square for image fusion is well studied before. What is the novelty you bring in? From what I understood, you get the coefficients once from LR and once from HR scenarios. Then you subtract their difference from Pan? Is that the contribution of the paper?

A13: The explanation of contribution is added as:

“Unlike NUB_Pansharp, the PSD algorithm attempts to construct the relationship between a panchromatic image and a single multi-spectral band, thereby directly converting the panchromatic image into the multi-spectral band. But NUB_Pansharp attempts to construct the relationship between a panchromatic image and multiple multi-spectral bands, synthesizing degraded panchromatic image using multi-spectral bands, and enhance the spatial information of multi-spectral bands by coefficient modulation. ”

 Q14: L253-255: The 254 larger the R-square is, the smaller the RMSE is, and the closer the fitting result is…

A14: This sentence is modified as:

“Higher R-square and lower RMSE indicate better fitting goodness.”

 Reviewer 2 Report

In this paper a panchromatic and multi-spectral image fusion method based on the panchromatic spectral decomposition (PSD) is proposed. Particularly the paper is  focused   to solve the proportionality coefficient, the bias coefficient and the residual matrix using the low-resolution panchromatic and multi-spectral images. Experiments are carried out to demonstrate that the proposed  method permits to achieve fused image characterized by high colour fidelity and sharpness. Particularly they show that the PSD algorithm produce fused image with low colour distortion and high sharpness. In addition the  PSD algorithm results powerful to be applied to different sensors, including also high-resolution optical satellite sensors.

The paper presents a traditional comparison of different approaches for pan-sharpening: the proposed method is compared with  others including Brovey, GS, PCA, SFIM and Pansharp, where Brovey, GS, PCA are algorithm modules in ENVI, and Pansharp is algorithm module in PCI Geomatica.

The results and conclusions are sufficiently significant, but some aspects must be improved. I suggest:

To expand the result discussion before the conclusions.

To consider the possibility to introduce ERGAS to evaluate the quality of the fused images.

To insert more information about the used data Beijing-2.

To expand the number of references, above all on evaluation of data fusion result quality. I suggest to consider at least the papers listed below.

 M. Lillo‐Saavedra, C. Gonzalob, A. Arquerob and E. Martinez, “Fusion of multispectral and panchromatic satellite sensor imagery based on tailored filtering in the Fourier domain,”International Journal of Remote Sensing, vol. 26(6), pp. 1263-1268, Dic. 2004. DOI: 10.1080/01431160412331330239

Maglione, P., Parente, C., & Vallario, A. (2016). Pan-sharpening Worldview-2: IHS, Brovey and Zhang methods in comparison. Int. J. Eng. Technol, 8, 673-679.

Du, Q., Younan, N. H., King, R., & Shah, V. P. (2007). On the performance evaluation of pan-sharpening techniques. IEEE Geoscience and Remote Sensing Letters, 4(4), 518-522.

Jagalingam, P., & Hegde, A. V. (2015). A review of quality metrics for fused image. Aquatic Procedia, 4, 133-142.

Minor remarks:

Please, insert the units of image spatial resolution in table 1.

Please, supply indications about the localization of the study area, i.e. its extension in terms of geographical or plane coordinates (UTM-WGS84? WGS84?  …).

 Author Response

Dear Reviewer 2:

Thanks for your valuable advices. Our modify situation and reply are as follows:

 Q1: To expand the result discussion before the conclusions.

A1: The conclusions are modified as follows:

“This paper proposes a fusion method which can convert panchromatic band into high-resolution multi-spectral band directly by constructing the relationship between the two. Experiments show that the proposed method has good colour retention ability. Compared with NUB_Pansharp, RMSE decreased from 13.82 to 7.16, ERGAS decreased from 4.89 to 2.54, SNR increased from 15.41 to 21.13, and CC increased from 0.91 to 0.98. The proposed method has good adaptability to different sensors and features by testing multiple sets of data. The current problem of this algorithm is that the fusion performance depends on the correlation between the panchromatic band and the multi-spectral band. Band1~3 have strong correlation with panchromatic band, and the RMSE of their fusion results are 6.73, 6.59 and 8.17, respectively. Band4 has a weak correlation with panchromatic band, and the RMSE is increased to 18.94. How to improve the algorithm adaptation to Pan-MS correlation is a worthy direction in future work. ”

 Q2: To consider the possibility to introduce ERGAS to evaluate the quality of the fused images.

A2: As suggestion, the quality metrics are replaced as ERGAS, RMSE, SNR and CC.

 Q3: To insert more information about the used data Beijing-2.

A3: The description about Beijing-2 is added as:

“Beijing-2 is a constellation constituted by three push-broom optical remote sensing satellites, providing panchromatic images with 0.8m (Pan: 440~510nm) and multi-spectral images (B1: 450~510nm, B2: 510~590nm, B3: 600~670nm, B4: 760~910nm).”

 Q4.To expand the number of references, above all on evaluation of data fusion result quality. I      suggest to consider at least the papers listed below.

M. Lillo‐Saavedra, C. Gonzalob, A. Arquerob and E. Martinez, “Fusion of multispectral and panchromatic satellite sensor imagery based on tailored filtering in the Fourier domain,”International Journal of Remote Sensing, vol. 26(6), pp. 1263-1268, Dic. 2004. DOI: 10.1080/01431160412331330239

Maglione, P., Parente, C., & Vallario, A. (2016). Pan-sharpening Worldview-2: IHS, Brovey and Zhang methods in comparison. Int. J. Eng. Technol8, 673-679.

Du, Q., Younan, N. H., King, R., & Shah, V. P. (2007). On the performance evaluation of pan-sharpening techniques. IEEE Geoscience and Remote Sensing Letters4(4), 518-522.

Jagalingam, P., & Hegde, A. V. (2015). A review of quality metrics for fused image. Aquatic Procedia4, 133-142.

A4: The evaluation parameters are changed to RMSE, SNR, ERGAS and CC, and the above papers are added in references.

 Q5: Please, insert the units of image spatial resolution in table 1.

A5: The units of image spatial resolution in table 1 are inserted as suggested.

 Q6: Please, supply indications about the localization of the study area, i.e. its extension in terms of geographical or plane coordinates (UTM-WGS84? WGS84?  …).

A6: SuperView-1 and Ziyuan-3 don’t have coordinate information. So we only add the coordinate of Beijing-2 datum in article: “The study area is Kunming, China, with coordinates of 102.6176E and 25.1246N.”

 Round  2

Reviewer 1 Report

Thanks for the paper overhaul. It looks better now. However, it needs to be clearly reviewed by an editor fluent in English. I am afraid the paper does not qualify to be published in the current status. Plus, I have a few more comments:

Figure 1: what dataset do the images belong to. It is not a common practice to use images before introducing them. What are bands 1 to 4. Why the distribution of bands 1-3 is pretty linear while 4 is not.

UNB-Pansharp, not NUB_Pansharp!

Referring to the following conversation:

“Q13: L227: up to this line, I still don’t understand what the contribution of the paper is. Using least square for image fusion is well studied before. What is the novelty you bring in? From what I understood, you get the coefficients once from LR and once from HR scenarios. Then you subtract their difference from Pan? Is that the contribution of the paper?

A13: The explanation of contribution is added as:

“Unlike NUB_Pansharp, the PSD algorithm attempts to construct the relationship between a panchromatic image and a single multi-spectral band, thereby directly converting the panchromatic image into the multi-spectral band. But NUB_Pansharp attempts to construct the relationship between a panchromatic image and multiple multi-spectral bands, synthesizing degraded panchromatic image using multi-spectral bands, and enhance the spatial information of multi-spectral bands by coefficient modulation. ””

I am not still convinced that this is your contribution. You are just comparing your method to UNB_Pansharp. What is your contribution? Your contribution needs to go to the introduction clearly explaining what is missing from the literature and what di authors add to the work to make it better.

“and points out that the theoretical basis is only approximation in practice” what does this mean?

Figure 3, again what dataset?

L230: how did you generate E_HR? how did you generate an RMS for it? I don’t need the formula for how to calculate RMS. What I ask is what is your original value and what is your estimated value from the difference of which you get the RMS.

Figure 4: what dataset again? Please make sure to refer to the dataset for every figure you add to your paper.

table 4, instead of ‘/’ use “NA” or ‘-‘

Author Response

Dear Reviewer:

Thanks for your careful reviews and valuable comments. The modifications are as follows:

 Q1: Figure 1: what dataset do the images belong to. It is not a common practice to use images before introducing them. What are bands 1 to 4. Why the distribution of bands 1-3 is pretty linear while 4 is not.

A1: Introduction of the dataset in Figure 1 and the explanation of this phenomenon have been added as follows:

“The correctness of formula (1) is based on that there is zero-bias linear relationship between panchromatic and multi-spectral images. However, this premise is not accurate enough in practice. A set of Beijing-2 data is taken as an example. Beijing-2 has a panchromatic band (Pan: 450~650nm) and four multi-spectral bands (B1: 440~510nm, B2: 510~590nm, B3: 600~670nm, B4: 760~910nm). Figure 1 records the digital number (DN) distribution of the panchromatic image and its corresponding multi-spectral bands. The distribution of bands 1~3 is pretty linear than band4. That is because the spectral ranges of bands 1~3 have a certain overlap with the panchromatic band, but band 4 does not.”

 Q2:UNB-Pansharp, not NUB_Pansharp!

A2: These errors are modified in this version.

 Q3:Referring to the following conversation:

“Q13: L227: up to this line, I still don’t understand what the contribution of the paper is. Using least square for image fusion is well studied before. What is the novelty you bring in? From what I understood, you get the coefficients once from LR and once from HR scenarios. Then you subtract their difference from Pan? Is that the contribution of the paper?

A13: The explanation of contribution is added as:

“Unlike NUB_Pansharp, the PSD algorithm attempts to construct the relationship between a panchromatic image and a single multi-spectral band, thereby directly converting the panchromatic image into the multi-spectral band. But NUB_Pansharp attempts to construct the relationship between a panchromatic image and multiple multi-spectral bands, synthesizing degraded panchromatic image using multi-spectral bands, and enhance the spatial information of multi-spectral bands by coefficient modulation. ””

I am not still convinced that this is your contribution. You are just comparing your method to UNB_Pansharp. What is your contribution? Your contribution needs to go to the introduction clearly explaining what is missing from the literature and what di authors add to the work to make it better. 

A3: This part is modified as:

“Compared with the existing ratio transformation methods, the PSD algorithm adds the bias term and the residual term into the Pan-MS relationship model, which enhances the accuracy. In addition, the ratio transformation methods will add the same high-frequency information into different bands, which is an important cause of color distortion. The PSD algorithm generates the fused image by converting the panchromatic image directly, which can avoid this problem effectively.”

 Q4:“and points out that the theoretical basis is only approximation in practice” what does this mean?

A4: This sentence is replaced by the follows:

This paper first analyses the ratio transformation methods, and points out that a zero-bias relationship between panchromatic and multi-spectral images is the premise of these methods. However, it is inaccurate to describe the two with a zero-bias linear relationship. There are bias and residual terms between the two in practice.

 Q5: Figure 3, again what dataset?

A5: The introduction of Figure 3 is added as:

“The simulated Beijing-2 data is applied for linear fitting analysis. Figure 3 records the relationship between the panchromatic band and the multi-spectral band 1 before and after the residual matrix elimination.”

 Q6:L230: how did you generate E_HR? how did you generate an RMS for it? I don’t need the formula for how to calculate RMS. What I ask is what is your original value and what is your estimated value from the difference of which you get the RMS.

A6: The data used in Section 3.3 is the simulation data of Beijing-2 which has been introduced in Section 3.1. The simulation data include PAN_HR, MS_HR and MS_LR. The parameters k and b can been obtained by linear fitting. So the E_HR can be obtained by PAN_HR, MS_HR, k and b:

E_HR = PAN_HR-k*MS_HR-b

()

PAN_LR can be obtained by down-sampled. So the estimated error matrix (the up-sampled E_LR) can be obtained by PAN_LR, MS_LR, k and b:

Up-sampled E_LR = Up-sampled [PAN_LR – k* MS_LR-b]

E_HR and the up-sampled E_LR are matrices of the same size, ant they can be subtracted directly. So their difference is still a matrix, and the RMS of it is calculated by normal.

 Q7:Figure 4: what dataset again? Please make sure to refer to the dataset for every figure you add to your paper.

A7: The introduction of Figure 4 is added as:

“The simulated Beijing-2 data is applied for residual matrix analysis. Figure 4 presents a set of visual residual matrices”

 Q8:table 4, instead of ‘/’ use “NA” or ‘-‘

A8: Modified as suggested

 In addition, this manuscript has been polished by American Journal Experts (AJE).